# Assessment of Saudi Parents’ Beliefs and Behaviors towards Management of Child Fever in Saudi Arabia—A Cross-Sectional Study

**DOI:** 10.3390/ijerph18105217

**Published:** 2021-05-14

**Authors:** Mohamed N. Al Arifi, Abdulrahman Alwhaibi

**Affiliations:** Department of Clinical Pharmacy, College of Pharmacy, King Saud University, Riyadh 11451, Saudi Arabia; aalwhaibi@KSU.EDU.SA

**Keywords:** Saudi parents, beliefs and behaviors, fever, Saudi Arabia

## Abstract

Objective: Fever alone can lead to rare serious complications in children, such as febrile seizures. The aim of this study is to assess the knowledge, beliefs, and behavior of parents toward fever and its management. Methods: A cross-sectional study using an online questionnaire was applied over a period of 3 months, from January to March 2018, to parents who were living in Saudi Arabia. The inclusion criteria were a parent who is a resident of Saudi Arabia, with at least one child aged 6 years or less, while incomplete questionnaires, having a child aged more than 6 years, or parents who were not living in Saudi Arabia were excluded. Results: A total of 656 parents completed the questionnaire. More than two-thirds of the subjects were female, the majority of whom were aged between 25–33 years old. The best-reported place to measure the temperature of children was the armpit (46%), followed by the ear (28%) and the mouth (10.7%). More than half of the parents considered their children feverish at a temperature of 38 °C. The majority of parents (79.7%) reported that the most serious side effects of fever were seizure, brain damage (39.3%), coma (29.9%), dehydration (29.7%), and death (25%). The most common method used to measure a child’s temperature was an electronic thermometer (62.3%). The most common antipyretic was paracetamol (84.5%). Conclusions: Our study demonstrates the good knowledge of parents in identifying a feverish temperature using the recommended route and tools for measuring body temperature.

## 1. Introduction

Despite that fever alone can lead to rare serious complications in children, such as febrile seizures, it is considered a self-manageable condition that can be managed at home with nonpharmacological or pharmacological (over-the-counter) medications [1]. However, overloaded visits to emergency medicine departments, pediatricians, and primary care offices are noticeable, possibly due to the vast spread of “fever phobia” phenomena among parents and healthcare providers [2,3,4,5,6,7,8,9,10]. The concept of “fever phobia” was first introduced in 1980 to detail parents’ spurious fright of fever and to highlight erroneous beliefs of its management [11]. The fear and misconception of fever were not solely restricted to parents. It includes other healthcare providers [12,13]. While earlier reports have revealed that fever has positive effects and is a sign of the significant improvement of the immune system, negative perceptions, like fears of febrile seizures, have caused health fever phobia, which remains challenging. Physicians continue to reduce low-grade fever without other symptoms and recommend various kinds of antipyretics to feverish children as initial treatment [5,14]. Several health professional agencies have published guidelines for healthcare providers and parents to instruct them on how to detect, manage, and monitor fever in children in order to halt the unrealistic fear of fever [15,16]. However, numerous reports from parents and other healthcare providers have ensured the insufficient knowledge, inappropriate attitude, misleading behaviors, and false beliefs of fever in children in different communities [4,5,6,7,8,9,10,11,12,13].

A previous study by AlAteeq et al. among Saudi parents evaluated the knowledge and practice in the home management of fever and indicated poor knowledge and practice with regards to child fever. In addition to this previous finding, the authors concluded the overuse of nonprescribed fever medication and the possible waste of health resources [17]. Similarly, another study by Hussain et al. indicated many misconceptions regarding fever among the majority of parents. In addition to this, Saudi parents have demonstrated undue fear of consequent body damage from fever and also believe that antibiotics can reduce high temperature [18]. One of the ultimate goals of any local healthcare system is to provide appropriate care for children and enough educational knowledge for their parents, in the same manner as international and other healthcare systems, about the most common illnesses, such as fever. To accomplish such a goal and to release evident-based instructions that comply with different societies, the assessment of the knowledge, attitude, and beliefs of the public toward fever and its management is required. Although several studies in different communities over the years have demonstrated the unrealistic fear and misconception of fever, no similar study has been conducted in Saudi Arabia [6,7,10,19]. The aim of this study is to assess the knowledge, beliefs, and behavior of parents toward fever and its management.

## 2. Materials and Methods

A cross-sectional study using an online questionnaire was applied. It was carried out during the period of January to March 2018 among Saudi parents who were living in Saudi Arabia. The inclusion criteria were a parent who is a resident of Saudi Arabia, living in Saudi during the time, with at least one child aged 6 years or less; exclusion criteria were incomplete questionnaires, having a child aged more than 6 years, and parents who were not living in Saudi Arabia at the time.

### 2.1. Development and Validation and Content of Questionnaires

A survey questionnaire was developed based on previous studies to assess the knowledge, attitude, and practice of parents towards fever in children and its management (KAP) [17,18,20]. The KAP survey contained questions about the methods utilized for measuring body temperature, frequency of checking the temperature, attitude towards the complications of fever, methods to select the appropriate drug or doses administered, attitude towards alternating drugs, and practices in obtaining and using antibiotic drugs. In addition, this study used questions to assess the parents’ sociodemographic data. The questionnaire was translated to the Arabic language by an independent professional translator using the forward–back translation procedure [21]; an expert in the field was requested to comment independently on the suitability of the questions in order to assess the validity of the questionnaire. After that, the questionnaire was validated using 10 randomly selected parents; the reliability test was determined through Cronbach’s alpha, and it was found to be 0.72.

### 2.2. Sample Size and Population

The sample size for this study was calculated using an online sample size calculator (http://www.raosoft.com/samplesize.html, accessed on 13 May 2021) by assuming a larger population size with a margin of error of ±5% and a confidence level of 95%, which resulted in a sample of 384 individuals [22].

### 2.3. Ethical Consideration

The present investigation was conducted according to the guidelines of the Checklist for Reporting Results of Internet E-Surveys (CHERRIES) [23]. In the survey questionnaires, there was a short paragraph stating the objective of the research and its importance and a statement on participation in the research being mandatory and participants having the full right to withdraw from the study at any point in time. Participants were also assured that their data would be used only for research purposes and would be considered confidential. Participants were also given confidence that there was no risk associated with participation in this study. Additionally, participants’ informed consent was obtained prior to answering the survey questions; they were also requested to provide authentic answers.

#### Data Extraction and Management

Data extraction is a crucial step in the research process and involves careful examination of completed and incompletely answered questionnaires [24]. For the current study, the data were checked for accuracy and completeness. Missing responses, incomplete responses, and invalid responses were excluded from the study, as shown in Figure 1.

### 2.4. Data Analysis

Descriptive statistics, including percentages, means, and frequency distributions, were used for each variable. Statistical Package for Social Sciences version 22.0 (SPSS Inc., Chicago, IL, USA) was used for statistical computations.

## 3. Results

In this study, a total of 656 parents completed the questionnaire. More than two-thirds of the subjects were female, with a majority of parents’ age ranging from 25 to 33 years. The youngest child of about 21% of the parents surveyed was one year of age. Most of the parents had a university education. The details of the demographic data are summarized in Table 1.

### 3.1. Attitude of Parents towards Fever and Its Management

The most common place to measure the temperature of children was the armpit (46%), followed by the ear (28%), mouth (10.7%), and then by hands or feet (6.1%). In this surveyed study, nearly 47% of parents thought that 36 °C was the normal body temperature. More than half of parents considered their children feverish at a temperature of 38 °C. In this study, only 20% of parents stated using the rectal route for medication; however, the reasons for using the rectal route were its high efficacy (16.2%), child refusal (12%), the belief of it being more practical (6.3%), and the presence of vomiting (5.3%).

The majority of parents (79.7%) reported that the most serious side effects of fever were seizure, followed by brain damage (39.3%), coma (29.9%), dehydration (29.7%), and death (25%), as presented in Table 2.

### 3.2. Knowledge of Parents of Methods in Managing Fever

In this study, the most common method used to measure a child’s temperature was an electronic thermometer (62.3%), followed by hands (32.2%), and a mercury-in-glass thermometer (3.5%) (Table 3). Additionally, this study assessed the knowledge of parents on drug therapy and other remedies to treat fever (Table 3). However, the most common antipyretics were paracetamol (84.5%) followed by antibiotics (19.4%) and ibuprofen (10.5%). On the other hand, the majority of parents (78.4%) also used cold-sponging to treat fever. To determine the appropriate dose of antipyretic medication, about 77% of parents stated that they normally use the spoon syringe provided with the drug for specific measurement.

### 3.3. Practice of Parents in Fever Treatment

Table 4 shows the practice of parents regarding fever and its treatment. About 60% of parents used pharmacological or nonpharmacological therapy if the temperature was 38 °C. In addition, 46 % of parents stated that they only called the doctor if the child’s temperature reached 39 °C, while 37.3% of parents called the doctor for a temperature of 38 °C.

This study found that the antipyretic drug selection was based on previous prescriptions for the same child (52.7%), followed by pharmacists’ recommendations (19.2%) and pediatricians’ recommendations (14.5%). In order to calculate the appropriate dose of the antipyretic drug to be administered to the feverish child, around 40.7% of parents would give a dose based on the previously prescribed dose, information in the package leaflet (33.5%), or a pharmacist’s recommendation (13%) (Table 4).

### 3.4. Attitude of Parents towards Antibiotic Use for Fever

Approximately 14.2 % of participating parents reported using antibiotics when their child had a fever, while 17.7% would use it when they suspected infection. Regardless of the reason for utilizing antibiotics, either for fever or suspected infection, about 13% of parents would obtain antibiotics without a physician’s prescription. Some parents (11.7%) thought that antibiotics should be prescribed to all children who develop fever, and 7.9% would be insistent on prescribing some antibiotics to their feverish children. Table 5 details the parents’ attitude towards antibiotic use for feverish children.

## 4. Discussion

Although there is no consensus on a single value to define fever since it differs based on the route of measurement, the thresholds of ≥38 °C for babies younger than 3 months and ≥39 °C for children who are 3 months and older are considered the feverish temperature [16,20]. Almost 92% of our participants believed that a temperature between 36–37.9 °C is the normal body temperature, and around 79.2% reported the fever temperature correctly at 38–39 °C. These results are in contrast to 34.3% of Australian parents who reported the fever temperature correctly at 38–39 °C, while our results seem comparable with caregivers in other communities [9]. The correct reporting of fever temperature among Jordanian, Greek, and Turkish parents were 78.25%, 66.1%, and 57.8%, respectively [7,20,25]. Two other local studies that focused solely on the parents who visited the emergency room or clinics for their febrile children revealed that the knowledge of the correct degree of fever had risen from 46% to 64% from 2000 to 2015 [25,26]. The high knowledge of fever temperature among parents in Saudi Arabia and the surrounding regions, specifically in recent years, may be the result of the frequent incident attacks of viruses causing fever, such as dengue fever, swine or avian influenza, or Middle East respiratory syndrome (MERS).

According to the recommendations, a digital thermometer device is the preferred tool to measure children’s core temperature [14,26,27]. The axillary method is the preferred measurement for children <4 weeks and all children, while the tympanic method seems to be the second preferred measurement for children ≥4 weeks. The oral measurement of temperature is not feasible in children because it is influenced by many factors, such food or liquid intake and patient cooperation for measurement [26,27,28]. The rectal method is not recommended to be used by caregivers due to its (1) invasive application, (2) slow detection in core temperature changes, (3) inaccuracy in the presence of blood or feces, and (4) risk of bacterial contamination or rectal perforation [29,30,31,32]. The results of the participating parents showed some consistency with national recommendations. Most of our participants used an electronic thermometer (62.3%) to measure body temperature; however, a slightly high number of parents (32.2%) still used their hands for fever detection. The tympanic method (46%) and the axillary (precisely in armpits area) method (28%) were the most-reported methods for fever detection, while the oral (10.7%) and rectal (3.4%) methods were the least-reported methods. In comparison with other caregivers in different communities, temperature-taking in the armpits of children was the commonly reported method among Greek (95.4%), Turkish (91.2%), Italian (82%), Emirati (60%), and Jordanian (43.2%) caregivers [4,7,20,25,28]. Although parents used the most recommended route for measuring body temperature, the accuracy and technique of measuring children’s body temperature by parents is still an essential concern [33].

Regardless of the route of administration, acetaminophen and ibuprofen are the recommended antipyretics for feverish children [34]. Some societies prefer an oral administration over a rectal administration of acetaminophen due to delay and inconstant absorption and the inaccuracy of the dose [26,27]. Most guidelines use weight-based dosing of acetaminophen and ibuprofen for children with fever, and some of them emphasize utilizing only the measuring devices provided with medications [35,36]. Cold or tepid sponging is not recommended since its evidence of efficacy is limited, its action is not directed toward the heat center, and its adverse effects of shivering and discomfort have been reported. Acetaminophen was used by 84.5% of our participants, while only 10.5% of them used ibuprofen. The oral route (79.4%) was the preferred route of administration, and the measuring devices provided with medications (77%) were used frequently by parents to measure the oral dose. Although most of the participating parents used the recommend antipyretics with the preferred route and measurements, around 78.4% and 13.6% were using cold and tepid sponging, respectively. In addition, about 19.4% of the participants used antibiotics as their choice for treating fever in children. In general, the use of antibiotics in Saudi Arabia without prescription for different issues seems to be trending downwards, from 78.7% in 2014 to 43.4% in 2017 [37,38], likely due to extensive education on the harmful effects of using antibiotics without prescription and the implementation of fines and penalties for breaking the law, implemented by the Saudi Ministry of Health, that prevents the sales of antibiotics without prescription [39,40]. According to a recent study that assessed the self-medication of antibiotics among Saudis in 2017, almost 29% reported using antibiotics for fever, while, in our study, only 14.2% of parents were using antibiotics for feverish children; around 13% of them were purchasing antibiotics directly from the pharmacies [35,36]. This is comparable to Jordanian parents (14%) who use antibiotics for their feverish children [20]. Although the monetary fines and penalties have diminished the sales of antibiotics without prescription, education on the use of antibiotics should be directed towards the public since there are still some parents (7.9%) who are insistent on prescribing antibiotics to their child. Although the online-conducted survey could be a limitation since some segments of the population cannot be approached, it seems to be a convenient method due to the high literacy rate and high usage of social media and chat applications [34,35]. Our study is not focused on the public, and it is only restricted to parents who have visited clinics or emergency departments with their feverish children because the aim of the study was to investigate the knowledge, practice, attitude, and belief of parents towards fever, especially from a Saudi mother’s perspective.

## 5. Conclusions

Our study has demonstrated the good knowledge of parents in identifying feverish temperatures using the recommended route and tools for measuring body temperature and selecting the appropriate antipyretic. However, constructive easy-to-read guidance, directed to parents for the management of feverish children, should be formulated and implemented by local agencies with a focus on the use of preferred nonpharmacological interventions, ways of appropriate utilization of various measuring devices, and the avoidance of antibiotics as a management technique for fever alone.

## Figures and Tables

**Figure 1 ijerph-18-05217-f001:**
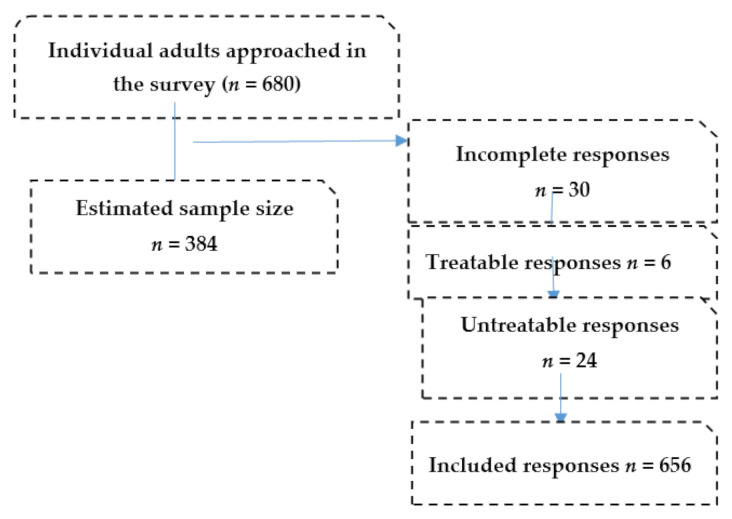
Flowchart of the responses.

**Table 1 ijerph-18-05217-t001:** Demographic data of respondents (*n* = 656).

Variables		Number	Percentages
Gender	Male	248	37.8
	Female	408	62.2
Age of parents	Less than 18	23	3.55
	18–24 years	152	23.2
	25–33 years	258	39.3
	34–51 years	215	32.8
	52–64 years	7	1.1
	More than 65	1	0.2
Age of youngest child	less than 1 year	115	17.5
	1	137	20.9
	2	109	16.6
	3	104	15.9
	4	92	14
	5	89	13.6
	6	10	1.5
Education level	Illiterate	5	0.8
	Primary/secondary school	28	4.3
	High school	144	22
	University	432	65.9
	Postgraduate	47	7.2
Insurance	None	266	40.5
	General	254	38.7
	Private	136	20.7

**Table 2 ijerph-18-05217-t002:** Attitude of parents towards fever and its management.

Variables		Number	Percentages
The most common places temperature is measured	The rectum	22	3.4
	The mouth	70	10.7
	The armpit	184	28
	The ear	302	46
	by hands or feet	40	6.1
	I don’t know	38	5.8
Opinions towards normal body temperature	36 °C	311	47.4
	37 °C	291	44.4
	38 °C	11	1.7
	39 °C	1	0.2
	40 °C	1	0.2
	41 °C	1	0.2
	I don’t know	40	6.1
Opinions towards fever temperature	36 °C	4	0.6
	37 °C	37	5.6
	38 °C	359	54.7
	39 °C	161	24.5
	40 °C	65	9.9
	41 °C	8	1.2
	I don’t know	22	3.4
Opinions about what temperature needs a dose of drugs	36 °C	6	0.9
	37 °C	55	8.4
	38 °C	392	59.8
	39 °C	156	23.8
	40 °C	26	4
	41 °C	4	0.6
	I don’t know	17	2.6
Reasons for favoring the administration of the medication rectally	More efficacy	106	16.2
	More practical	41	6.3
	Because of vomiting	35	5.3
	Because of child refusal	79	12
	According to a physician’s prescription	18	2.7
Complications of fever	Brain damage	258	39.3
	Seizure	523	79.7
	Death	164	25
	Coma	196	29.9
	Dehydration	195	29.7
	None	26	4

**Table 3 ijerph-18-05217-t003:** Knowledge of parents’ methods in managing childhood fever.

Variables		Number	Percentage
Method for measuring temperature	Hand	211	32.2
	Electronic thermometer	409	62.3
	Mercury-in-glass thermometer	23	3.5
	I do not check my child’s temperature	7	1.1
	I do not know	6	0.9
Frequency of measuring temperature, every	less than 15 min	83	12.7
	15–30 min	227	34.6
	30 min–1 h	169	25.8
	1–2 h	122	18.6
	More than 2 h	55	8.4
Drugs for fever	Acetaminophen	554	84.5
	Ibuprofen	69	10.5
	Aspirin	14	2.1
	Antibiotics	127	19.4
	Other	29	4.4
Remedies used in addition to drugs	Cold sponging	514	78.4
	Ice pack	5	0.8
	Tepid sponging	89	13.6
	I use drugs only	29	4.4
		19	2.9
Site of medication administration	Oral	521	79.4
	Rectal	135	20.6
Tools used to administer the medication	Teaspoon	53	8.1
	Specific measurement spoon or syringe provided with the drug	505	77
	Measuring spoon or syringe of other drugs	98	14.9

**Table 4 ijerph-18-05217-t004:** Practice of parents with regard to fever.

Variables		Number	Percentage
When treatment is administered	36 °C	6	0.9
	37 °C	55	8.4
	38 °C	392	59.8
	39 °C	156	23.8
	40 °C	26	4
	41 °C	4	0.6
	I don’t know	17	2.6
When doctor is called	38 °C	285	37.3
	39 °C	302	46
	40 °C	91	13.9
	41 °C	12	1.8
	42 °C	1	0.2
	43 °C	5	
The right antipyretic drug based on	Previous advice from the pediatrician	346	52.7
	Consultation of pharmacist	126	19.2
	Consultation of other persons	18	2.7
	Information collected by media	19	2.9
	I decide by myself	39	5.9
	I call my pediatrician	95	14.5
	Other	13	2
Calculation of the appropriate dose of the fever-lowering drug based on	Previous advice from the pediatrician	267	40.7
	Reading the package leaflet	220	33.5
	Consultation of the pharmacist	85	13
	Media	4	0.6
	I decide by myself	10	1.5
	I call my pediatrician	65	9.9
	Other	5	0.8
To give a fever-lowering drug, you consider	Age	346	52.7
	Sex	126	19.2
	Weight	18	2.7
	Height	19	2.9
	Severity of fever	39	5.9
	Severity of illness	95	14.5
	Nothing	13	2

**Table 5 ijerph-18-05217-t005:** Attitude of parents towards antibiotic use for fever.

Variables		Number	Percentage
Antibiotic use	Child has fever	93	14.2
	Suspect an infection	116	17.7
	Doctor’s prescription	389	59.3
	Advertisements	4	0.6
	All above	54	8.2
Purchase an antibiotic from pharmacy	Have prescription	514	78.4
	Myself	85	13
	Someone	14	2.1
	Internet	8	1.2
	Other	35	5.3
In general, would you give antibiotics to your unwell child without consulting a physician	Yes	101	15.4
	No	555	84.6
In general, would you be insistent in prescribing some antibiotics to your child	Yes	52	7.9
	No	604	92.1
In general, would you use antibiotics based on a pharmacist’s consultation?	Yes	309	47.1
	No	347	52.9
Do you think antibiotics should be prescribed to all children who develop a fever?	Yes	77	11.7
	No	579	88.3

## Data Availability

Data will be available upon request from the corresponding author of the study.

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
