# Peer review of "Assessment of Saudi Parents’ Beliefs and Behaviors towards Management of Child Fever in Saudi Arabia—A Cross-Sectional Study"

_ijerph, 2021, doi:10.3390/ijerph18105217_

Round 1

Reviewer 1 Report

Nicely written paper. In the introduction, considering providing more specific examples of "spurious fright." What is meant by "possible waste of health resources" in line 45 and "misconceptions regarding fever" in line 47. Providing examples will help support the significance of the research aim.

Was the KAP survey back translated to ensure quality of translation?

In section 3.1, line 128, consider clarifying "the best reported place to measure temperature." Are you describing what participants believe are the best places to measure temperature or the most common places temperature is measured?

The data analysis and discussion were thorough and thoughtful. 

Only 32% (12) citations were published within the last 5 years. 

Author Response

Reviewer 1

Nicely written paper. In the introduction, considering providing more specific examples of "spurious fright." What is meant by "possible waste of health resources" in line 45 and "misconceptions regarding fever" in line 47. Providing examples will help support the significance of the research aim.

Spurious fright meaning, false attacks of ever, it is well known fact that fever is common for everyone, irrespective of age, a fever is simply the body’s natural reaction to an infection. In fact, fevers are a good sign that the immune system is working properly. Many times, a fever will run its course without any need for medical attention. While some parents are quick to give their children fever-reducing medication, it’s actually better to let the fever run its course. Remember, a fever is your body’s way of fighting an infection. If your child is acting fine and seems relatively unbothered, you don’t need to give them any medication. Fevers from infections do not cause brain damage, even when they get up to the 104 – 106 range. The only time an elevated temperature can cause brain damage is when it reaches above 108 degrees, and this only happens when the air temperature is high – as is the case of children being left in hot cars.

Thank you for your valuable comment, possible waste of health resources means in the previous study reported visit of health centers, use of thermometers and medicines and all health care utilities for the control of ever, is called as health resources

Was the KAP survey back translated to ensure quality of translation?

Answer by author

Thank you for your valuable comment, yes the translation procedure was used forward and backward translation procedure

In section 3.1, line 128, consider clarifying "the best reported place to measure temperature." Are you describing what participants believe are the best places to measure temperature or the most common places temperature is measured?

My apologies, yes this question talks about the most common places to measure the child temperature

The data analysis and discussion were thorough and thoughtful. 

Thank you for your valuable comment

Only 32% (12) citations were published within the last 5 years. 

Dear editor and team Thank you for your valuable comment, yes there were lack of studies in Saudi Arabia.

Reviewer 2 Report

The aim of study is to assess the knowledge, beliefs, behavior of parents toward fever in their children and its management.
The study is well done and the results obtained are interesting.
Comments:
The study design is as fit for purpose as the number of cases studied. Authors should specify in the text if they have had the approval of the Ethics Committee and they should specify if the data have been collected ensuring the anonymity of parents and children.
The tables need a restyling because each row must be associated with the corresponding numbers which in the version I received are sometimes above or below the corresponding row.
The bibliography is updated, but must be reviewed, in particular # 16, # 17, # 18, # 19 # 24 because the references are reposted without respecting the rules of the Journal.

Author Response

Reviwer-2

The aim of study is to assess the knowledge, beliefs, behavior of parents toward fever in their children and its management.
The study is well done and the results obtained are interesting.

Dear editor and team, thank you very much for your valuable time to review and your supportive comment., we are glad to publish with you.
Comments

The study design is as fit for purpose as the number of cases studied. Authors should specify in the text if they have had the approval of the Ethics Committee and they should specify if the data have been collected ensuring the anonymity of parents and children.
The tables need a restyling because each row must be associated with the corresponding numbers which in the version I received are sometimes above or below the corresponding row.

Thank you very much or your valube comment, this study was exempted from ethical permission, the study titled of Assessment of Saudi mother’s beliefs and behaviors towards Childhood fever and its management. This study is descriptive, non-intervention cross sectional study that would be conducted through online without direct contact with the patient or public intervention. Free non-obligatory choice of participants to assure the online questionnaire. As a results this study was waived from IRB approval

attached is the approval letter 

The bibliography is updated, but must be reviewed, in particular # 16, # 17, # 18, # 19 # 24 because the references are reposted without respecting the rules of the Journal.

Dear editor and team, thank you very much for your valuable time to review, I have corrected it as per your comments. 

Round 2

Reviewer 1 Report

Thank you for your responses and edits. Please add a sentence about the back translation in the methods.

Author Response

Dear editor and team Thank you very, much for the comment, i have corrected asper your suggestion 

underlined in method section with red color 

The questionnaire was translated to Arabic language by an independent professional translator using forward-back translation procedure [22]